# Identifying Oxidized Lipid Metabolism-Related LncRNAs as Prognostic Biomarkers of Head and Neck Squamous Cell Carcinoma

**DOI:** 10.3390/jpm13030488

**Published:** 2023-03-08

**Authors:** Ziwei Zhang, Byeong Seop Kim, Wenqing Han, Xiaojun Chen, Yingjie Yan, Li Lin, Gang Chai

**Affiliations:** Department of Plastic and Reconstructive Surgery, Shanghai Ninth People’s Hospital, School of Medicine, Shanghai Jiao Tong University, Shanghai 200011, China

**Keywords:** oxidized lipid metabolism, lncRNAs, head and neck squamous cell carcinoma, prognostic, risk score

## Abstract

The relationship between oxidized lipid metabolism and the immunological function of cancer is well known. However, the functions and regulatory mechanisms of lncRNAs associated with oxidized lipid metabolism in head and neck squamous cell carcinoma (HNSCC) remain to be fully elucidated. In this study, we established an oxidized lipid metabolism-related lncRNA prognostic signature to assess the prognosis and immune infiltration of HNSCC patients. The HNSCC transcriptome was obtained from The Cancer Genome Atlas. The choice of the target genes with a relevance score greater than 10 was performed via a correlation analysis by GeneCards. Patients were categorized by risk score and generated with multivariate Cox regression, which was then validated and evaluated using the Kaplan–Meier analysis and time-dependent receiver operating characteristics (ROC). A nomogram was constructed by combining the risk score with the clinical data. We constructed a risk score with 24 oxidized lipid metabolism-related lncRNAs. The areas’ 1-, 2-, and 3-year OS under the ROC curve (AUC) were 0.765, 0.724, and 0.724, respectively. Furthermore, the nomogram clearly distinguished the survival probabilities of patients in high- and low-risk groups, between which substantial variations were revealed by immune infiltration analysis. The results supported the fact that oxidized lipid metabolism-related lncRNAs might predict prognoses and assist with differentiating amid differences in immune infiltration in HNSCC.

## 1. Introduction

Head and neck squamous cell carcinomas (HNSCCs) are common tumors that affect many tissues and organs, such as the nasal cavity, paranasal sinuses, nasopharynx, oral cavity, pharynx, and larynx [1]. HNSCCs rank eighth in the world in terms of morbidity and mortality, with about 500,000 people catching this disease each year and 350,000 dying from it [2,3]. HNSCC is damaging to both human health and quality of life due to its high local recurrence rate, high metastatic rate, and poor prognosis. Even though surgery, radiation, and chemotherapy are used frequently, the 5-year survival rate for people with HNSCC remains around 50%, with local recurrence rates close to 50% and distant metastasis rates over 25% [1]. Due to these problems, it is important to find new, sensitive prognostic markers for HNSCC tumors as soon as possible to reduce the number of HNSCC patients who are not diagnosed before their disease becomes worse.

Long noncoding RNAs (lncRNAs) are RNA transcription products that are longer than 200 nucleotides and which cannot code for proteins [4]. The presence of lncRNAs in the nucleus, especially in the chromatin fraction, demonstrates their regulatory role in gene transcription [5,6]. Previous research has shown that lncRNAs play many functions in the formation of HNSCC, including those associated with cancer growth, recurrence, and spread [1,7,8]. Recent studies suggest that lncRNAs’ influence not only extends to the architecture of the genome or transcriptome, but also to the lipid oxidation metabolism of the tumor microenvironment [9,10,11]. As such, this causes the principal cancer phenotypes.

Reactive oxygen species (ROS), such as superoxide anions, hydrogen peroxide, hydroxyl radicals, singlet oxygen, and lipid peroxyl radicals, are typical byproducts of several critical metabolic activities, and their creation is an inescapable consequence of aerobic life [12]. ROS have physiological roles in normal quantities but harmful effects at high concentrations on cells. Oxidative stress is the harm that occurs when a living organism produces an excessive amount of ROS [13,14,15]. Increased levels of ROS can harm macromolecules, such as DNA, proteins, and lipids in a way that is “non-specific”. Due to their high diffusibility, hydroxyl radicals attack DNA quickly, leading to the formation of DNA lesions, such as oxidized DNA bases, as well as single strand or double strand breaks. Proteins are primarily harmed by ROS through modifications to their amino acid residues, which, in turn, change their functions [13]. Moreover, it can lead to the oxidation of cell membrane polyunsaturated fatty acids (PUFAs) via free radical chain reactions, and the creation of lipid hydroperoxides as the main products [16]. Lipid peroxidation that is generated by oxidative stress is connected with human physiology and illness, including cancer [17].

Unsaturated fatty acid peroxidation is a well-known metabolic process that results in a complicated combination of volatile chemical molecules, such as those comprising aldehydes. Random lipid peroxidation happens when cancer cells have an excess of ROS, leading to the creation of different aldehydes [18]. The development of reactive aldehydes that target biological components and can launch a chain reaction is triggered by lipid peroxidation [19]. Recent animal and human research have demonstrated that the peroxidation of polyunsaturated fatty acids and the generation of lipid peroxidation products activate the apoptosis signaling system and inhibit the growth of cancer [20,21]. ROS causes lipid peroxidation and cell death by reacting with polyunsaturated fatty acids on lipid membranes. Previous studies have found that, in cancer cell lines lacking mtDNA, lipid peroxidation increases the permeability to hydrogen peroxide, leading to cell death [22]. Lipid peroxidation and the production of lipid ROS are also linked to iron toxicity in cancer cells [23]. In addition, using lipid peroxidation-mediated endoplasmic reticulum stress, docosahexaenoic acid monoglyceride causes apoptosis and autophagy in breast cancer cells [24]. Moreover, anticancer drugs, such as doxo-rubicin, depend on the stimulation of lipid peroxidation to exert anticancer effects [25].

Although it has been established that lncRNAs may act as biomarkers for HNSCC [1,8], the prognostic significance of oxidized lipid metabolism-related lncRNAs remains unclear. The goal of our study is to find out if oxidized lipid metabolism-related lncRNAs can be used to predict a patient’s prognosis, as well as to determine the degree of tumor infiltration in HNSCC patients.

## 2. Materials and Methods

### 2.1. Acquisition of Information of Patients with HNSCC

The Cancer Genome Atlas (TCGA, https://cancergenome.nih.gov, accessed on 1 February 2023) was accessed to obtain the RNA sequencing data sets, as well as determine the matching clinical features of patients with HNSCC. Then, 44 samples of normal tissue and 504 samples of HNSCC were acquired.

Patients with HNSCC who possessed comprehensive lncRNA expression data, clinical information, and a follow-up period of at least 30 days were the inclusion criteria. Patients combined with other tumors or with less than 30 days of follow-up were excluded.

This research did not have to acquire ethical approval for our study because we used information that was already in the public database.

### 2.2. Selection of Oxidized Lipid Metabolism-Related LncRNAs

Oxidized lipid metabolism-related genes were downloaded from the GeneCards database (https://www.genecards.org/, accessed on 1 February 2023). On this database, the oxidized lipid metabolism-related genes with a relevance score greater than 10 were chosen. Patients’ LncRNA expression profiles were collected by TCGA. All data were normalized before further analysis with the limma package of the R (v.4.2.1, R Foundation for Statistical Computing, Vienna, Austria) software. Using R software, Pearson correlation studies were conducted on lncRNAs and autophagic genes in HNSCC patients (v.4.2.1). For autophagic lncRNAs, a correlation coefficient (R) > 0.4 and a *p*-value of 0.001 were considered significant. Using Cytoscape, a co-expression network between autophagic lncRNAs and genes was also constructed (v.3.9.1).

### 2.3. Establishment and Validation of the Risk Score

Using the clinical data of HNSCC cases from the TCGA, a univariate Cox proportional hazard regression analysis was utilized to select the lncRNAs associated with survival from oxidized lipid metabolism-related lncRNAs (*p* < 0.05). Then, we executed the LASSO regression with a 10-fold cross-validation and a value of 0.05, as well as ran it for 1000 cycles. In order to prevent overfitting, 1000 instances of random stimulation were performed per cycle. Then, a multivariate Cox regression analysis was employed to select the target genes and to develop an independent prognostic signature. The risk score for each patient sample was computed by multiplying the expression level of each lncRNA by the total of their weights in the multivariable Cox model. Patients were separated into high- and low-risk groups based on the median risk score. To validate the predictive value of the risk score, the Kaplan–Meier log-rank test, t-SNE, and ROC curve analysis were performed.

### 2.4. Validation of Prognostic Signature

On the basis of the clinical data, univariate and multivariate Cox regression analyses were conducted to establish if the risk score was an independent indication of prognosis. The link between prognosis and risk factors such as age, gender, grade, stage, tumor size (T), lymph node metastasis (N), and risk score was examined using the c-index and ROC curve analysis.

### 2.5. Construction and Application of Nomogram

Using the rms R package (v.4.2.1, R Foundation for Statistical Computing, Vienna, Austria), the risk score, age, gender, grade, tumor size (T), lymph node metastasis (N), and tumor stage were utilized to establish a nomogram and a calibration curve in order to evaluate whether the predicted outcome exhibited a high degree of concordance with the observed data. To validate the predictive usefulness of the nomogram in different clinical stages, the Kaplan–Meier log-rank test was used.

### 2.6. Functional Enrichment Analysis

Functional enrichment was assessed using the Kyoto Encyclopedia of Genes and Genomes (KEGG), Gene Ontology (GO), and gene set enrichment analysis (GSEA; v.4.2.3; http://www.broadinstitute.org/gsea/index.jsp, accessed on 1 February 2023).

### 2.7. Analysis of Immune Cell Infiltration in High- and Low-Risk Patient Groups

The gene expression matrix data were screened and analyzed using CIBERSORT (https://cibersortx.stanford.edu/) [26]. CIBERSORT was utilized to infer the relative proportion of the 22 immune cells invading each supplement sample. To evaluate the association between the lncRNA signature and immune cell infiltration, the infiltrating immune cell populations from the high- and low-risk groups were compared.

The ssGSEA in the R Bioconductor package (v.4.2.1, R Foundation for Statistical Computing, Vienna, Austria) of the Gene Set Variation Analysis (GSVA), with default parameters, was used to estimate the extent of infiltration of the various immune cell groups [27]. The ssGSEA algorithm is a rank-based approach that calculates a score expressing the degree of absolute enrichment of a gene set in each sample. The ssGSEA algorithm was fed the gene sets from other publications.

### 2.8. Statistical Analysis

Except for the gene set enrichment analysis, all statistical analyses in this study were carried out using the R software (v.4.2.1, R Foundation for Statistical Computing, Vienna, Austria). Unless otherwise specified, *p* < 0.05 was regarded as statistically significant.

## 3. Results

### 3.1. Identification of Oxidized Lipid Metabolism-Related LncRNAs

From The Cancer Genome Atlas (TCGA), we obtained 44 normal samples and 504 tumor samples. A total of 7912 oxidized lipid metabolism-related genes were downloaded from GeneCards (https://www.genecards.org/, accessed on 1 February 2023). The heat map below displays the expression of 50 genes (random selection) between the normal group and the tumor group (Figure 1A). According to the expression of the 1252 oxidized lipid metabolism-related genes (relevance score > 10) between the normal and tumor samples, we finally obtained 323 different expression mRNAs. Of them, 196 were up-regulated, and the others were down-regulated (Figure 1B). The differentially expressed genes are also shown in box plots (Figure 1C). 

The results of KEGG and GO enrichment analysis among the differential genes are displayed in Figure 2.

### 3.2. Establishment and Verification of the 24-LncRNAs Prognostic Risk Scores

Through correlation analysis, a total of 2817 oxidized lipid metabolism-related lncRNAs met the criteria (R > 0.4 and *p* < 0.001) (Figure 3A). Finally, 24 genes were filtered for the construction of risk scores by univariate Cox regression, LASSO regression (Figure 3B,C), and multivariate Cox regression analysis. Based on the median risk scores, 249 and 244 HNSCC patients were classified in either high- or low-risk groups. The heatmap was used to show the expression of these lncRNAs between the high- and low-risk groups (Figure 3D). The Kaplan–Meier curve analysis showed that the OS of each group was statistically different, with the high-risk group having a worse OS (Figure 3E).

The results of the ROC curve indicated that the risk scores significantly discriminated the HNSCC patients’ outcomes at 1, 3, and 5 years (Figure 3F). Based on the risk scores, we classified the HNSCC patients in order of their risk ratings (Figure 3G). The scatter diagram showed a link between the HNSCC patients’ survival rates and their risk scores. As the risk score went up, the death rate went up (Figure 3H). The PCA and t-SNE analyses were undertaken to assess the expression differences between the low- and high-risk groups across all patients (Figure 4A,D,G), train groups (Figure 4B,E,H), and test groups (Figure 4C,F,I). The results of the PCA showed the risk scores based on the 24 oxidized lipid metabolism-related lncRNAs could clearly distinguish the high-risk patients from the low-risk patients across all patients, train groups, and test groups. The results of the KEGG and GO enrichment analyses among the differential genes between low- and high-risk groups are displayed in Figure 5. We constructed a co-expression network of 24 oxidized lipid metabolism-related lncRNAs that comprise the risk score (Figure 6A). In addition, we found that 9 lncRNAs (AC009159.1, AC013472.2, AC016747.2, AC027237.2, AC091729.2, AC119396.1, AC138356.1, AL121672.3, and AL135937.1) were the protective factors, whereas 15 lncRNAs (AC004832.5, AC017048.3, AC092115.3, AC116036.2, AL078644.1, CASC8, CDC42-IT1, CDKN2A-DT, GRHL3-AS1, LINC01088, LINC01506, LINC01572, SNHG16, UVRAG-DT, and Z97205.1) were the risk factors in the Sankey diagram (Figure 6B). To investigate the differences in biological functions between the high- and low-risk groups, we utilized GSEA software to explore the high-risk group (Figure 6C) and low-risk group (Figure 6D) in the KEGG pathway (top 10) in the entire set. 

### 3.3. Assessment of Nomogram

The results of the univariate and multivariate Cox regression indicated that age, lymph node metastasis (N), and risk score are independent risk factors (HR > 1, *p* < 0.05) (Figure 7A,B). The nomogram was based on the risk score, age, gender, grade, tumor size (T), lymph node metastasis (N), and tumor stage, as is shown in Figure 7C. In addition, we performed 1-, 3-, and 5-year calibration plots to demonstrate that the nomogram accurately predicted the 1-, 3-, and 5-year OS (Figure 7D). To test the usefulness of the nomogram, we also conducted tumor-stage-based subgroup analyses (stage I-II vs. stage III-IV; Figure 7E,F). The results of the C-index and ROC analysis indicated that the risk score can predict the patient prognosis more precisely (Figure 7G,H).

### 3.4. Immune Cell Infiltration

First, the analysis of immunological function is shown in a heatmap (Figure 8A). In addition, the composition of the 22 kinds of immune cells in each sample is presented in a histogram (Figure 8B) and violin plot (Figure 8C). The immune cells that were associated with survival are presented in Figure 8D–H. Notably, the infiltration of naive B cells, resting mast cells, activated T cells with CD4 memory, and T cells with follicular helpers were discovered to be significantly correlated with a longer survival time. By contrast, the activated mast cells suggested a poor prognosis. The differences in the immune cell and immune function scoring between low- and high-risk groups are displayed as violin plots (Figure 8I,J).

## 4. Discussion

This study investigated the involvement of the lncRNAs and genes involved in oxidative lipid metabolism in HNSCC. We validated a signature of 24 lncRNAs that were associated with oxidative lipid metabolism as a predictive biomarker for HNSCC, dividing patients into high- and low-risk groups. In addition, it was also revealed by immune cell infiltration that the infiltration of naive B cells, resting mast cells, activated T cells with CD4 memory, and T cells with follicular helpers was substantially associated with a longer survival period. In contrast, active mast cells were indicative of a dismal prognosis.

Head and neck cancers are among the most globally prevalent malignancies, with squamous cell carcinomas accounting for around 90% of cases [28]. The local control rate and quality of life of HNSCC patients have improved as a result of developments in surgical techniques. In early stage HNSCC, surgery coupled with chemoradiotherapy yields an excellent prognosis [2]. However, early stage HNSCC patients are typically asymptomatic. The majority of patients are identified in the middle or late stages, and approximately 17% miss the surgical window [29]. The initial symptoms, which frequently result in delayed diagnoses and mimic common illnesses, include nasal congestion, oral ulcers, sore throats, and hoarseness. The site of origin affects the clinical presentation. Further, 10% of patients initially present with distant metastases that are stage IVC, and over 40% of patients have regional nodal involvements and diseases classified as stage IVA or B [2,30]. A 5-year survival rate for patients with this illness ranges between 40% and 50% [1]. 

Due to advances in sequencing technology, genome-wide profiling has revealed that about 98% of transcriptional outputs are lncRNAs. The recent discovery of lncRNA involvement in numerous biological activities has created a new situation for better understanding complicated processes such as cancer [1,7,8]. Excitingly, mounting data indicates that lncRNAs play a crucial regulatory function in the growth of HNSCC tumors [2,31,32]. For instance, lncRNA lnc-POP1-1, which is increased by VN1R5, interacts with MCM5 to induce cisplatin resistance in HNSCC [7]. LncRNA LINC00460 induces EMT in HNSCC by enhancing the nuclear import of peroxiredoxin-1 [8]. Even more surprising is that M1 macrophage-derived exosomes and their key molecule, lncRNA HOTTIP, stop the growth of HNSCC by turning up the TLR5/NF-kappaB pathway [1]. 

Despite the fact that lncRNAs have been discovered as biomarkers for HNSCC, their prognostic value in connection to oxidative lipid metabolism remains unclear. To date, however, only 4 of 24 lncRNAs’ functions (LINC01572, LINC01088, CASC8, and SNHG16) were introduced by previous research. The overexpression of LINC01572 significantly increased hepatocellular carcinoma (HCC) cell proliferation, migration, invasion, and epithelial-mesenchymal transition (EMT), whereas the knockdown of LINC01572 had the opposite impact on HCC cells [33]. Additionally, LINC01572 is involved in tumor treatment resistance. As a competing endogenous RNA for miR-497-5p, LINC01572 decreased miR-497-inhibitory 5p’s impact on ATG14, resulting in chemo-induced autophagy and the chemoresistance of gastric cancer (GC) cells [34]. These results demonstrate that the LINC01572 gene is a risk factor for tumor prognosis. Our results also indicated that LINC01572 is a risk factor of HNSCC patients’ outcome. Previous research has provided some insight into the regulatory function of LINC01088 in tumors. By suppressing p21 by binding to EZH2, LINC01088 can enhance cell proliferation, ultimately aggravating the tumorigenicity of non-small cell lung cancer (NSCLC) [35]. LINC01088 directly targets miR-548b-5p and miR-548c-5p, as well as enhances the expression of G3BP1 and PD-L1, consequently boosting the development of colorectal cancer and immune evasion [36]. In addition, among central nervous system tumors, LINC01088 exerts pro-oncogenic activities in glioma via binding to SNRPA and regulating SNRPA mRNA transcription [37]. A previous study has proven the effect of the CASC8 rs1562430 polymorphism on esophageal squamous cell carcinoma (ESCC) susceptibility and discovered that functional polymorphisms in CASC8 rs1562430 A > G may influence an individual’s susceptibility to ESCC [38]. The USP21/YY1/SNHG16 axis is involved in the proliferation, migration, and invasion of non-small-cell lung cancer (NSCLC) [9]. Moreover, highly expressed SNHG16 is an oncogenic lncRNA that targets miR-146a to enhance tumor growth and poor prognosis in NSCLC [11]. The level of SNHG16 was high in pancreatic cancer (PC) samples and cell lines, and its levels were linked to poor differentiation, advanced TNM, lymph node metastasis, and poor overall survival [39]. In colorectal cancer (CRC), the reduced expression of SNHG16 impacts the genes that are involved in lipid metabolism [40]. 

According to the results of the co-expression network, these lncRNAs may regulate seven oxidative lipid metabolism-related genes ANXA5 [41], SERPINE1 [42], CYP3A5 [43], GSTA2 [44], IGF1R [45], ACLY [46], and PYCR1 [47]. Through the ERK/Nrf2 pathway, ANXA5 might protect testicular Leydig and Sertoli cells from DBP-induced oxidative stress damage [41]. Furthermore, MiR-506-3p may exacerbate oxidative stress in testicles in vivo and in vitro by inhibiting the expression of ANXA5, and by decreasing the activity of the Nrf2/HO-1 pathway [48]. The level of ANXA5 correlates with in vitro and in vivo tumor aggressiveness and lymphatic metastasis in mouse hepatocarcinoma cells [49]. Through the ERK2/c-Jun/p-c-Jun (Ser73) and ERK2/E-cadherin pathways, ANXA5 promotes the malignancy of murine hepatocarcinoma Hca-F cells in vitro. It is a key molecule in metastasis, especially LNM, and it may be a good place to start treating hepatocellular cancer [50]. However, up to now, there have been no reports on ANXA5 and HNSCC. Previous research has indicated that SERPINE1 plays a crucial role in controlling the origin and development of HNSCC and might be identified as critical indicators for the accurate diagnosis and prognosis of HNSCC; hence, providing prospective therapeutic targets [51]. Consistent with its relationship with a poor prognosis in HNSCC patients, SERPINE1 overexpression enhances tumor aggressiveness and metastasis to lymph nodes and the lung [52]. In addition, functional experiments targeting SERPINE1 expression have indicated that suppressing SERPINE1 expression inhibited the malignant phenotype of HNSCC cells [53]. The results of this paper showed LINC01572 is a risk factor in HNSCC. In addition, it may affect the prognosis of HNSCC patients by regulating SERPINE1. By regulating mTORC2/Akt Signaling, CYP3A5 serves as a tumor suppressor in hepatocellular carcinoma [43]. In addition, by promoting androgen receptor (AR) nuclear translocation, CYP3A5 governs the proliferation of prostate cancer cells [53]. The abovementioned findings show that the CYP3A5 may be a component that inhibits tumor growth. Nonetheless, investigations on the CYP3A5 in HNSCC have not yet been revealed. IGF1R down-regulation inhibits the growth of tumor cells and renders HNSCC cells to be more susceptible to cisplatin in the laboratory [54]. According to the results of the co-expression network, we inferred that SNHG16 may regulate the expression of IGF1R.

ACLY is a crucial enzyme that connects gluconeogenesis with lipid metabolism. Increased glucose absorption and quicker glycolytic flux result in an increase in mitochondrial citrate synthesis in tumor cells with an altered energy metabolism. The pharmacological or genetic suppression of ACLY dramatically decreases cancer cell growth and promotes apoptosis. ACLY has been demonstrated to be aberrantly expressed in a variety of tumor types [55]. Some researchers believe that ACLY influences the repair of DNA damage and is a predictor of the outcome of radiation for HNSCC [56]. SNHG16 was associated with ACLY in the analysis of co-expression network. However, the regulating role between the two must be experimentally examined further. Finally, no research on PYCR1 and GSTA2 in HNSCC have been published; hence, they will not be discussed further in this article.

This study had multiple limitations. First, HNSCC is a catch-all term for multiple distinct cancers that should be studied individually. The current model relies on a publicly available database; therefore, it requires validation via using more extensive data. In addition, more experiments are needed to clarify the underlying molecular mechanisms and to identify the more promising therapeutic avenues.

## 5. Conclusions

Our findings highlight the potential prognostic value of the 24 lncRNA signatures involved in the oxidative lipid metabolism for HNSCC, thereby allowing for the risk stratification of patients—which could enlighten clinical decisions and treatment plans in order to improve outcomes.

## Figures and Tables

**Figure 1 jpm-13-00488-f001:**
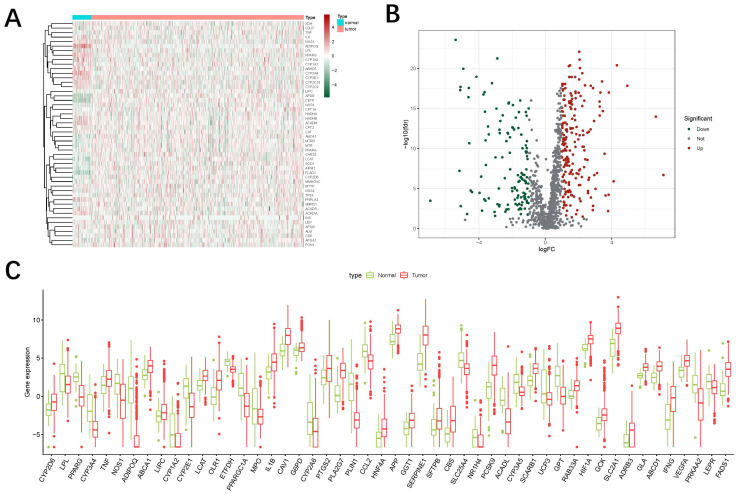
Expression of genes related to oxidative lipid metabolism between HNSCC and normal tissues. (**A**) Heat map showing the expression levels of oxidized lipid metabolism-related genes in two samples. Green represents low expression and red represents high expression. (**B**) Volcano map showing differentially expressed lncRNAs associated with oxidative lipid metabolism in the two samples. Red dots represent up-regulated genes, green dots represent down-regulated genes, and black dots represent unchanged genes. (**C**) Boxplot showing the differentially expressed lncRNAs associated with oxidative lipid metabolism in the two samples.

**Figure 2 jpm-13-00488-f002:**
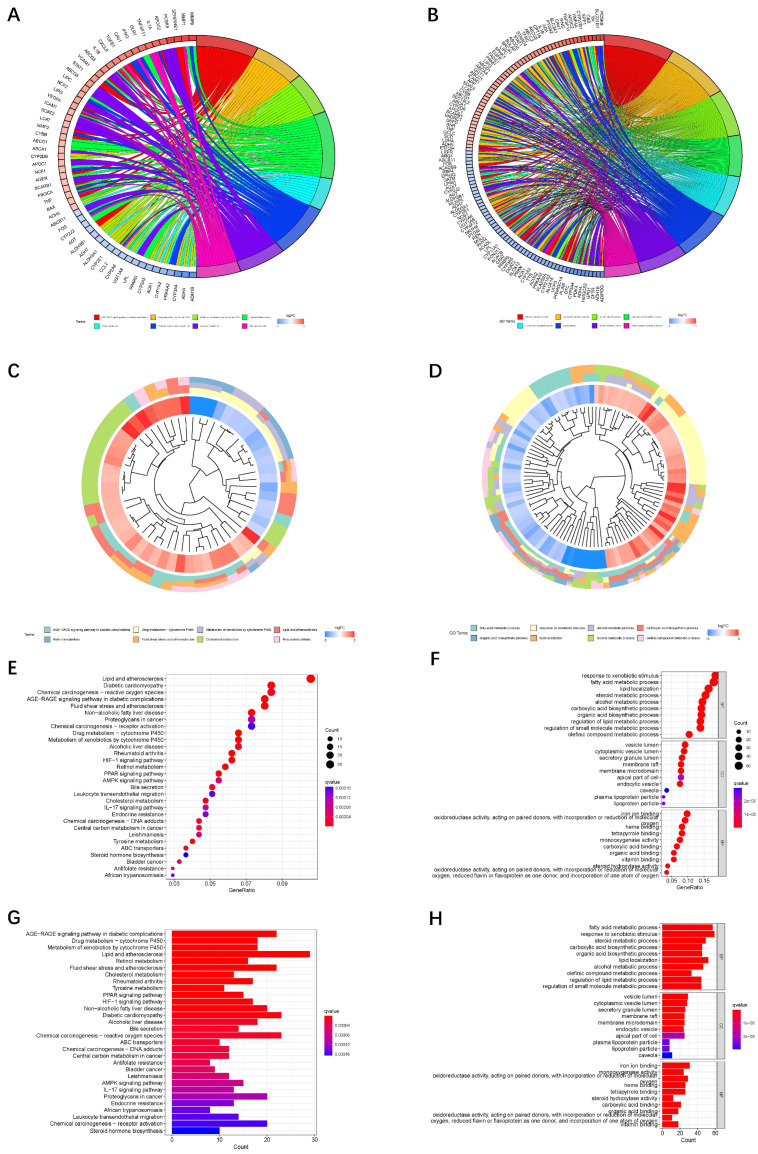
KEEG and GO analyses of the differentially expressed genes between two samples. (**A**,**C**,**E**,**G**): KEGG analysis of the differentially expressed genes in two samples. (**B**,**D**,**F**,**H**): GO analysis of the differentially expressed genes in two samples.

**Figure 3 jpm-13-00488-f003:**
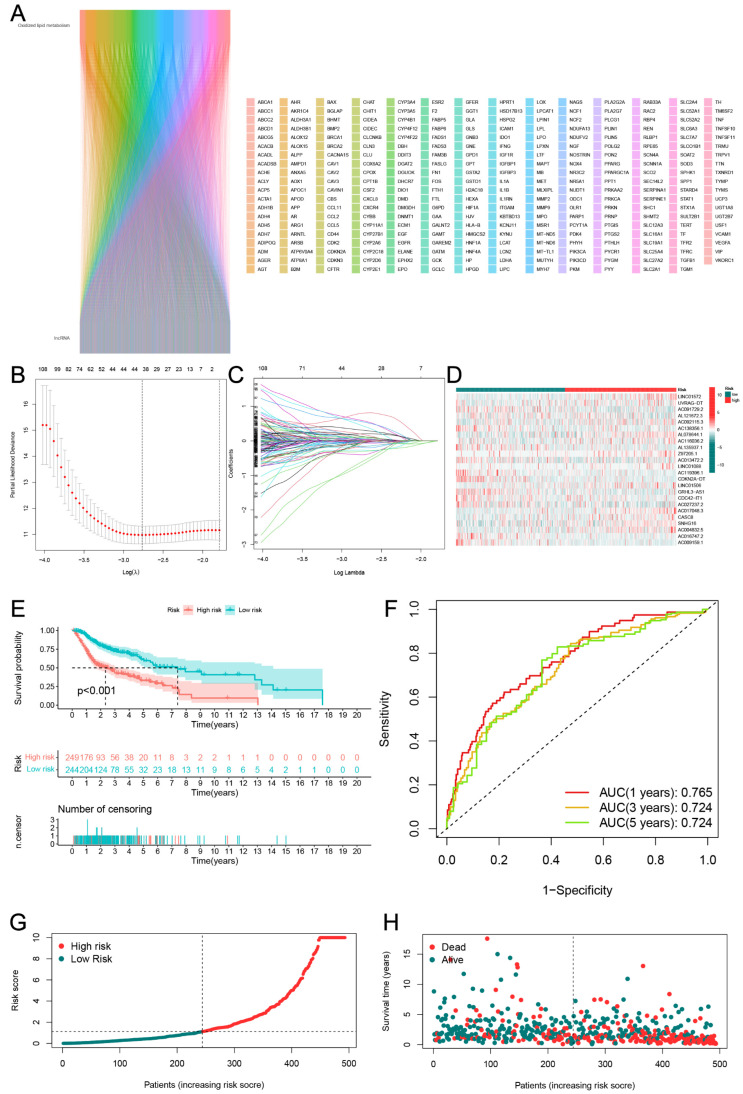
Identification of the oxidative lipid metabolism-related lncRNAs associated with prognosis in HNSCC. (**A**) Prognostic lncRNA extracted by univariate Cox proportional hazard regression analysis. (**B**,**C**) LASSO-Cox regression model for screening the process of oxidative lipid metabolism-related lncRNAs. (**D**) Heat map showing the expression of 24 lncRNAs between the high-risk and low-risk groups. (**E**) Kaplan–Meier curves for the survival status and survival time in the high-risk and low-risk groups for HNSCC. (**F**) The ROC curves show the predictive efficiency of the risk scores for patient survival at 1, 3, and 5 years. (**G**,**H**) The distribution of risk scores, survival status, and survival time in patients with HNSCC.

**Figure 4 jpm-13-00488-f004:**
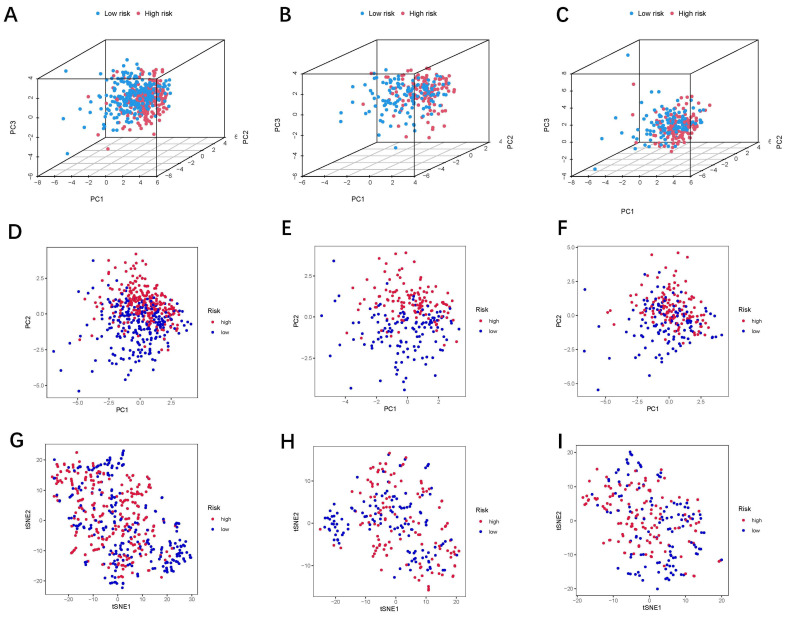
The PCA and t-SNE plot analysis between the low-risk and high-risk groups, based on the 24 oxidative lipid metabolism-related lncRNAs expression profiles. (**A**–**C**) PCA-3D plot analysis for all patients, train, and test groups. (**D**–**F**) PCA plot analysis for all patients, train, and test groups. (**G**–**I**): tSNE plot analysis for all patients, train and test groups.

**Figure 5 jpm-13-00488-f005:**
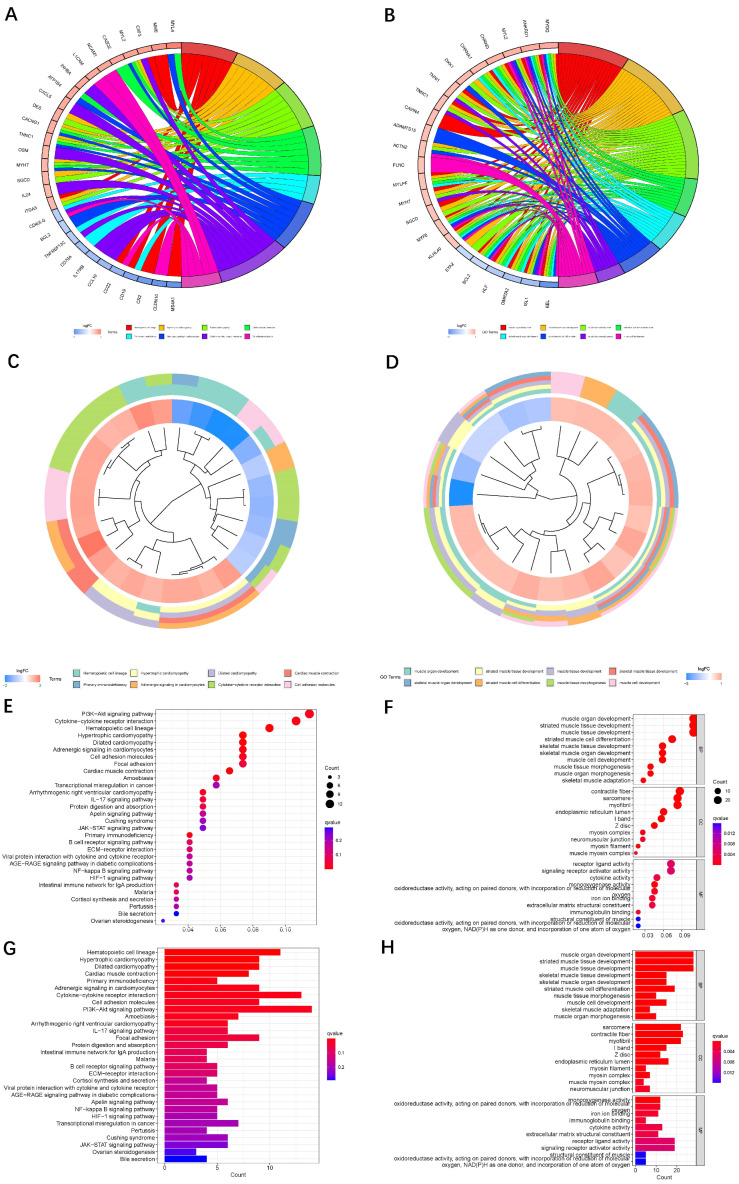
The KEEG and GO analyses of differential genes between the low- and high-risk groups. (**A**,**C**,**E**,**G**): KEGG analysis of the differentially expressed genes in two groups. (**B**,**D**,**F**,**H**): GO analysis of the differentially expressed genes in two groups.

**Figure 6 jpm-13-00488-f006:**
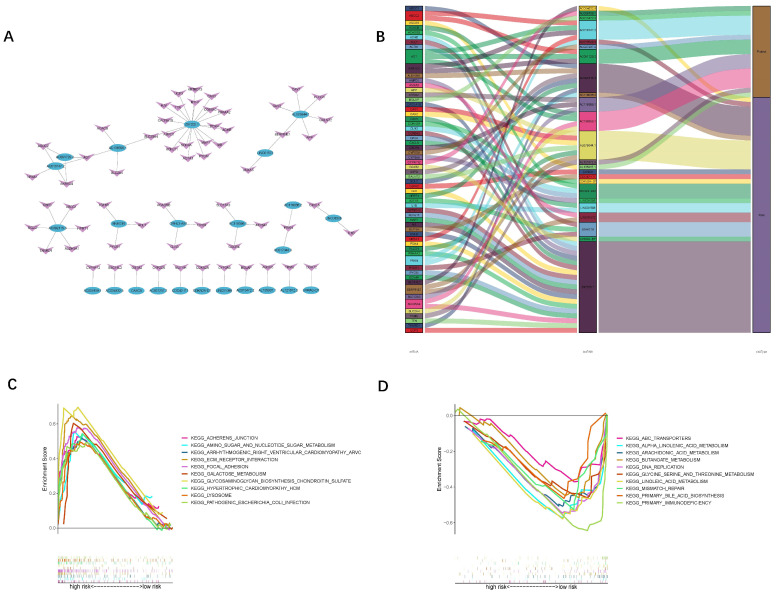
(**A**) The lncRNAs–mRNA co-expression network of HNSCC. (**B**) The Sankey diagram shows the relationship between the oxidative lipid metabolism-related mRNAs and the 24 lncRNAs. (**C**) The top 10 pathways significantly enriched in the high-risk group of the GSEA analysis. (**D**) The top 10 pathways significantly enriched in the low-risk group of the GSEA analysis.

**Figure 7 jpm-13-00488-f007:**
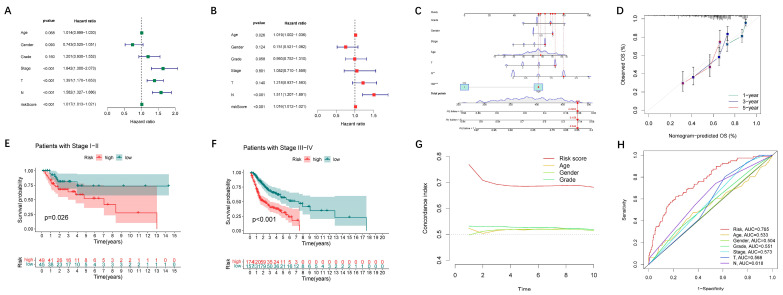
Assessment of the prognostic risk model of the 24 oxidative lipid metabolism-related lncRNAs in HNSCC. (**A**,**B**) The univariate and multivariate Cox regression analysis of the risk scores and clinical characteristics for prognostic value. (**C**) The nomogram model is based on a risk score, age, gender, grade, tumor size (T), lymph node metastasis (N), and tumor stage. (**D**) The calibration plots are for the 1-, 3-, and 5-year overall survival. (**E**,**F**) The Kaplan–Meier curves are for the tumor stage-based subgroup analyses. (**G**) The C-index is for the risk scores, grade, age, and gender. (**H**) ROC curves for the risk scores and the clinical characteristics for predicting prognosis.

**Figure 8 jpm-13-00488-f008:**
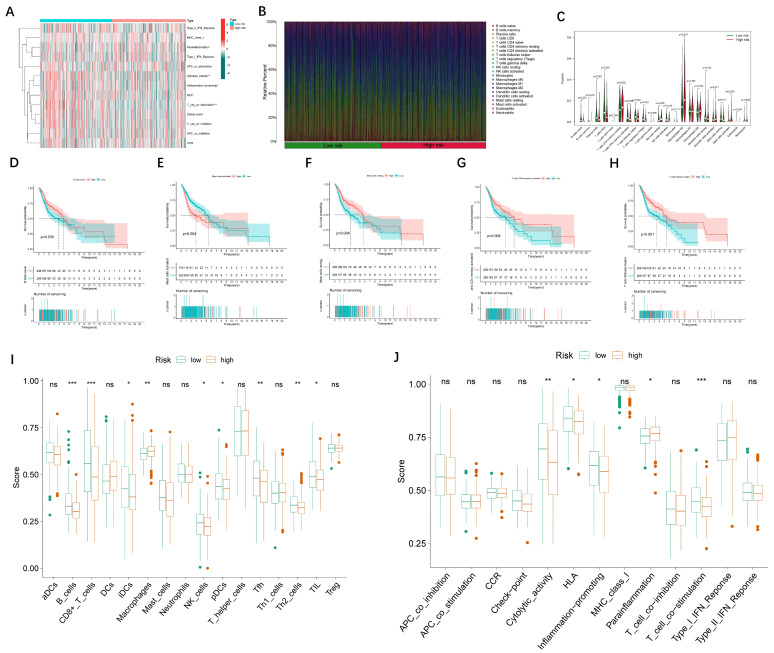
The analysis of immune infiltration in high-risk and low-risk groups. (**A**) A heat map of the differential immune cells in two groups. (**B**) A histogram showing the proportion of immune cells in the two groups. (**C**) A violin plot showing the difference in the level of immune cell infiltration between the two groups. (**D**–**H**) Kaplan–Meier curves showing the immune infiltrating cells that are associated with prognosis. (**I**,**J**) A violin diagram showing differences in immune cell and immune function scores between the two groups. * *p* < 0.05, ** *p* < 0.01, *** *p* < 0.001, ns: no significance.

## Data Availability

The datasets generated and/or analyzed during the current study are publicly available in the TCGA.

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
