# Peer review of "Identifying Oxidized Lipid Metabolism-Related LncRNAs as Prognostic Biomarkers of Head and Neck Squamous Cell Carcinoma"

_jpm, 2023, doi:10.3390/jpm13030488_

Round 1

Reviewer 1 Report

Dear authors,

Thank you for conducting this study entitled "Identifying Oxidized Lipid Metabolism-Related LncRNAs as Prognostic Biomarkers of Head and Neck Squamous Cell Carcinoma". I appreciate your great effort. I have a few comments to improve the presentation of the article:

1.      Please take care of using the abbreviations like HNSCC and ROS throughout the whole manuscript because you wrote the full terms of them when they appeared first.

2.      This sentence (line 77) "Although it has been established that lncRNAs may act as biomarkers for HNSCC," needs a reference.

3.      The authors should mention the exclusion criteria.

4.      Please take care that the p < 0.05 or 5%  (not at 0.05 as in line 142) is a statistically significant difference.

5.      Ethical approval from your IRB should be provided.

6.      Discussion

a.       "Despite the fact that lncRNAs have been discovered as biomarkers for HNSCC, their prognostic value in connection to oxidative lipid metabolism remains unclear. This study investigated the involvement of lncRNAs and genes involved in oxidative lipid metabolism in HNSCC to discover oxidative lipid metabolism-related lncRNAs. This was followed by the validation of a signature of 24 lncRNAs associated with oxidative lipid metabolism as a predictive biomarker for HNSCC. The signature was utilized to divide HNSCC patients into high- and low-risk groups. In addition, the results of immune cell infiltration showed a significant difference between high- and low-risk groups. It was also revealed that the infiltration of naive B cells, resting mast cells, activated T cells with CD4 memory, and T cells with follicular helper was substantially associated with a longer survival period. In contrast, active mast cells were indicative of a dismal prognosis." This paragraph is a repetition of the introduction, methods, and results. I advise the authors to put the strength or the main outcome of the study in the first paragraph.

b.      There are no limitations to the study.

7.      I see that the conclusion is a repetition of the objectives and methods. Therefore, you must rewrite it depending on your results.

Author Response

Response to Reviewer 1 Comments

Point 1: Please take care of using the abbreviations like HNSCC and ROS throughout the whole manuscript because you wrote the full terms of them when they appeared first.

Response 1: We are grateful for the reviewer's suggestion. We have changed the full terms to corresponding abbreviation throughout the manuscript except for their first occurrences to increase brevity.

Changes in the text:

Our document has been modified using Microsoft Word Track Changes. If you do not see any changes, click on the Review menu in Microsoft Word and select Final Showing Markup (or All Markup). Please also ensure that there is a check mark next to 'Insertions and Deletions' in the Show Markup dropdown menu.

  The manuscript has been modified accordingly.

Point 2: This sentence (line 77) "Although it has been established that lncRNAs may act as biomarkers for HNSCC," needs a reference.

Response 2: We are grateful for the reviewer's comment. We have added the appropriate references and marked them in the text.

Changes in the text:

Our document has been modified using Microsoft Word Track Changes. If you do not see any changes, click on the Review menu in Microsoft Word and select Final Showing Markup (or All Markup). Please also ensure that there is a check mark next to 'Insertions and Deletions' in the Show Markup dropdown menu.

  This sentence is in line 83 in the revised manuscript, and the references have been marked out as shown.

Point 3: The authors should mention the exclusion criteria.

Response 3: We are grateful for the reviewer's comment. We have added a description of the exclusion criteria in Materials and Methods serction to make the criteria for obtaining data more rigorous.

Changes in the text:

Our document has been modified using Microsoft Word Track Changes. If you do not see any changes, click on the Review menu in Microsoft Word and select Final Showing Markup (or All Markup). Please also ensure that there is a check mark next to 'Insertions and Deletions' in the Show Markup dropdown menu.

  Additions are in line 94-95.

Point 4: Please take care that the p < 0.05 or 5%  (not at 0.05 as in line 142) is a statistically significant difference.

Response 4: Thanks to the reviewer for the reminder. We apologize for the typographical mistake that caused the content error. We have added the missing symbol.

Changes in the text:

Our document has been modified using Microsoft Word Track Changes. If you do not see any changes, click on the Review menu in Microsoft Word and select Final Showing Markup (or All Markup). Please also ensure that there is a check mark next to 'Insertions and Deletions' in the Show Markup dropdown menu.

  The missing “<” has been added in the text (line 149).

Point 5: Ethical approval from your IRB should be provided.

Response 5: We are grateful for the reviewer's suggestion. Since we were using data that was already available to the general public, approval from the ethical boards to conduct our research was not required.

Changes in the text:

Our document has been modified using Microsoft Word Track Changes. If you do not see any changes, click on the Review menu in Microsoft Word and select Final Showing Markup (or All Markup). Please also ensure that there is a check mark next to 'Insertions and Deletions' in the Show Markup dropdown menu.

  None.

Point 6a: Discussion: "Despite the fact that lncRNAs have been discovered as biomarkers for HNSCC, their prognostic value in connection to oxidative lipid metabolism remains unclear. This study investigated the involvement of lncRNAs and genes involved in oxidative lipid metabolism in HNSCC to discover oxidative lipid metabolism-related lncRNAs. This was followed by the validation of a signature of 24 lncRNAs associated with oxidative lipid metabolism as a predictive biomarker for HNSCC. The signature was utilized to divide HNSCC patients into high- and low-risk groups. In addition, the results of immune cell infiltration showed a significant difference between high- and low-risk groups. It was also revealed that the infiltration of naive B cells, resting mast cells, activated T cells with CD4 memory, and T cells with follicular helper was substantially associated with a longer survival period. In contrast, active mast cells were indicative of a dismal prognosis."This paragraph is a repetition of the introduction, methods, and results. I advise the authors to put the strength or the main outcome of the study in the first paragraph.

Response 6a: We are grateful for the reviewer's comment. We recapitulated the important results of this paper and highlighted them in the first paragraph of the discussion.

Changes in the text:

Our document has been modified using Microsoft Word Track Changes. If you do not see any changes, click on the Review menu in Microsoft Word and select Final Showing Markup (or All Markup). Please also ensure that there is a check mark next to 'Insertions and Deletions' in the Show Markup dropdown menu.

  We have modified the manuscript accordingly. Relevent content was adjusted to the first paragraph of the Discussion section (line 268-279).

Point 6b: Discussion: There are no limitations to the study.

Response 6b: Thank you for your comments about the limitations. After careful consideration and review, we added the limitations at the end of our discussion. We believe that the limitations are mainly related to the disease type itself and the study design.

Changes in the text:

Our document has been modified using Microsoft Word Track Changes. If you do not see any changes, click on the Review menu in Microsoft Word and select Final Showing Markup (or All Markup). Please also ensure that there is a check mark next to 'Insertions and Deletions' in the Show Markup dropdown menu.

  Discussions about the limitations were in line 388-392.

Point 7: I see that the conclusion is a repetition of the objectives and methods. Therefore, you must rewrite it depending on your results.

Response 7: We are grateful for the reviewer's comment. We have improved the language and removed the repetitive words. We also added the potential clinical significance of the results of this study.

Changes in the text:

Our document has been modified using Microsoft Word Track Changes. If you do not see any changes, click on the Review menu in Microsoft Word and select Final Showing Markup (or All Markup). Please also ensure that there is a check mark next to 'Insertions and Deletions' in the Show Markup dropdown menu.

  The conclusion section has been rewritten (line 394-400).

Reviewer 2 Report

This is a relevant topic that will expand knowledge about oxidized lipid metabolism-related LncRNAs. The manuscript is well-written and featuresan adequate Introduction as well as Material and Methods description. In the Introduction we suggest the authorsdescribe other effects induced by reactive oxxygen species such mitochondrial dysfunction, DNA fragmentation and degradation of the sulfhydryl groups of the proteins that make up the cytoskeleton.

The analyzes reinforced the evidence strength of the results.

I suggest to adjust the formatting of the figures because some words in the graphics are difficult to read.

The authors also must include a paragraph in the Discussion that discusses the morphological characteristics of HNSCC as Inc RNAs expression may reflect differences when considering whether the tumor is well-differentiated, moderately or undifferentiated.

Author Response

Response to Reviewer 2 Comments

Point 1: In the Introduction we suggest the authors describe other effects induced by reactive oxygen species such mitochondrial dysfunction, DNA fragmentation and degradation of the sulfhydryl groups of the proteins that make up the cytoskeleton.

Response 1: We are grateful for the reviewer's suggestion. For the damage of different cellular components caused by ROS, we have briefly added in the Introduction section. Corresponding references were also marked.

Changes in the text:

Our document has been modified using Microsoft Word Track Changes. If you do not see any changes, click on the Review menu in Microsoft Word and select Final Showing Markup (or All Markup). Please also ensure that there is a check mark next to 'Insertions and Deletions' in the Show Markup dropdown menu.

  The manuscript has been modified accordingly in line 55-61.

Point 2: The analyzes reinforced the evidence strength of the results. I suggest to adjust the formatting of the figures because some words in the graphics are difficult to read.

Response 2: We are grateful for the reviewer's comment. We have uploaded larger resolution images in the revised manuscript and refined the figure captions.

Changes in the text:

Our document has been modified using Microsoft Word Track Changes. If you do not see any changes, click on the Review menu in Microsoft Word and select Final Showing Markup (or All Markup). Please also ensure that there is a check mark next to 'Insertions and Deletions' in the Show Markup dropdown menu.

  Clearer figures were uploaded, with more detailed descriptions in the figure captions below.

Point 3: The authors also must include a paragraph in the Discussion that discusses the morphological characteristics of HNSCC as Inc RNAs expression may reflect differences when considering whether the tumor is well-differentiated, moderately or undifferentiated.

Response 3: We are grateful for the reviewer's comment. We have provided some demonstration regarding the clinical characteristics of HNSCC in the first paragraph of the Discussion section.

Changes in the text:

Our document has been modified using Microsoft Word Track Changes. If you do not see any changes, click on the Review menu in Microsoft Word and select Final Showing Markup (or All Markup). Please also ensure that there is a check mark next to 'Insertions and Deletions' in the Show Markup dropdown menu.

  We have modified the manuscript accordingly, and corresponding discussion was in line 286-291.
